# Deep Learning for Sex Estimation from Whole-Foot X-Rays: Benchmarking CNNs for Rapid Forensic Identification

**DOI:** 10.3390/diagnostics15222923

**Published:** 2025-11-19

**Authors:** Rukiye Çiftçi, İpek Atik, Özgür Eken, Monira I. Aldhahi

**Affiliations:** 1Department of Anatomy, Medical Faculty, Gaziantep Islam Science and Technology University, Gaziantep 28010, Turkey; rukiye.ciftci@gibtu.edu.tr; 2Department of Electrical Electronics Engineering, Gaziantep Islam Science and Technology University, Gaziantep 27010, Turkey; ipek.atik@gibtu.edu.tr; 3Department of Physical Education and Sport Teaching, Faculty of Sports Sciences, Inonu University, Malatya 44280, Turkey; ozgur.eken@inonu.edu.tr; 4Department of Rehabilitation Sciences, College of Health and Rehabilitation Sciences, Princess Nourah bint Abdulrahman University, P.O. Box 84428, Riyadh 11671, Saudi Arabia

**Keywords:** sex estimation, foot radiographs, forensic applications, convolutional neural networks (CNNs), foot, deep learning benchmarking

## Abstract

**Background:** Accurate sex estimation is crucial in forensic identification when DNA analysis is impractical or remains are fragmented. Traditional anthropometric approaches often rely on single bone measurements and yield moderate levels of accuracy. **Objective:** This study aimed to evaluate deep convolutional neural networks (CNNs) for automated sex estimation using entire foot radiographs, an approach rarely explored. **Methods:** Digital foot radiographs from 471 adults (238 men, 233 women, aged 18–65 years) without deformities or prior surgery were retrospectively collected at a single tertiary center. Six CNN architectures (AlexNet, ResNet-18, ResNet-50, ShuffleNet, GoogleNet, and InceptionV3) were trained using transfer learning (70/15/15 train–validation–test split, data augmentation). The model performance was assessed using accuracy, sensitivity, specificity, precision, and F1-score. **Results:** InceptionV3 achieved the highest accuracy (97.1%), surpassing previously reported methods (typically 72–89%), with balanced sensitivity (97.5%) and specificity (96.8%). ResNet-50 followed closely (95.7%), whereas simpler networks, such as AlexNet, underperformed (90%). **Conclusions:** Deep learning applied to whole-foot radiographs delivers state-of-the-art accuracy for sex estimation, enabling rapid, reproducible, and cost-effective forensic identification when DNA analysis is delayed or unavailable, such as in mass disasters or clinical emergency settings.

## 1. Introduction

Sex estimation is a cornerstone of forensic identification, particularly when remains are incomplete or DNA analysis is impractical or delayed due to various reasons. Determining sex early narrows the pool of possible identities and facilitates subsequent profiling of age, stature, and ancestry [1,2,3,4]. Conventional approaches such as DNA or chromosomal testing can provide excellent accuracy but are costly, time-consuming, and may fail when biological material is degraded [1,5]. Therefore, forensic anthropologists frequently rely on skeletal analysis to achieve rapid and cost-effective sex estimation.

Bones of the pelvis and skull show the highest sexual dimorphism, but these structures are often fragmented or unavailable in mass disasters, trauma cases, or archaeological contexts [4,5]. In such situations, foot bones become practical alternatives: they are protected by footwear, relatively small and dense, and more resistant to taphonomic damage or post-mortem changes [6,7]. Numerous studies have explored sex estimation using individual foot elements such as the calcaneus and talus [4,6] or metatarsals and phalanges [7,8,9]. These works typically rely on morphometric measurements or classical machine-learning methods such as discriminant analysis or logistic regression, reporting accuracy rates between about 72–89% [7,10,11].

Radiologic imaging provides a reliable method for sex estimation. Radiologic methods, such as magnetic resonance imaging (MRI), computed tomography (CT), and X-ray imaging, are commonly used to analyze bone maturity and age. Among these imaging methods, X-ray is the most frequently preferred due to its simplicity, minimal radiation exposure, and multiple ossification centers [12].

Foot bones are among the bones frequently used in sex determination studies due to their presence in both forensic and archaeological contexts, their small and robust structure, their less exposure to taphonomic elements, their resistance to postmortem changes and their protection by foot [13].

With the rapid development of artificial intelligence, deep learning—especially convolutional neural networks (CNNs)—has transformed medical imaging by automatically learning complex patterns without manual feature extraction [14,15,16]. CNNs have shown strong performance in radiographic sex estimation of the hand, wrist, and other skeletal regions [1,10]. However, the use of entire foot radiographs with deep learning remains limited. Most available studies have focused on cropped bone regions or required manual landmarking, which limits automation and clinical applicability [7,8,9]. There is a clear unmet need for fully automated, high-accuracy models that can process standard digital foot X-rays already stored in picture archiving and communication systems (PACS).

This study addresses that gap by systematically benchmarking six CNN architectures—AlexNet, ResNet-18, ResNet-50, ShuffleNet, GoogleNet, and InceptionV3—for automated sex estimation using whole-foot radiographs. Unlike previous approaches, which are limited to single bones or handcrafted measurements, we trained and compared modern deep networks end-to-end on entire foot images.

We hypothesize that (1) deep CNN models applied to whole-foot radiographs will achieve substantially higher accuracy than previously reported morphometric or shallow machine-learning methods (typically 72–89%); [7,10], and (2) deeper architectures such as InceptionV3 and ResNet-50 will outperform shallower networks such as AlexNet, demonstrating the advantage of network depth for this complex forensic classification task.

## 2. Materials and Methods

### 2.1. Participants

The study was initiated with decision number 2022/9 from Bandırma Onyedi Eylül University Health Sciences Non-Interventional Ethics Committee. Images with good quality and no fractures in the foot region and ankle were included in the study, while images outside the desired age range, those with a history of foot surgery or fractures, and those with poor image quality were excluded from the study. A significance level (α) of 0.05, power (1 − β) of 0.80, effect size of 0.02, and actual power of 80.0 were used to perform a power analysis. The analysis indicated that at least 420 participants would be needed for the study [17]. Foot and ankle X-ray images of 471 individuals (238 males and 233 females) aged 18–65 who visited X University Training and Research Hospital between 1 January 2022, and 30 September 2024, were included in the study. Individuals who had undergone surgery for any reason, had bone fractures or subluxations, exhibited deformities, or degeneration were excluded from the sample.

### 2.2. Imaging Method

In this study, patient images obtained using a digital single-detector X-ray device (Jumong, SG Healthcare, Seoul, Republic of Korea) in the Radiology Department of X University Training and Research Hospital were used. All images in the sample were obtained from digital imaging and medical file format picture archiving and communication systems (DICOM) and transferred to medical imaging software (Horos Medical Image Viewer Version 3.0, United States). The transferred images were labeled as male or female.

This study proposes a deep learning approach for image-based classification problems. As shown in Figure 1, in the first phase, the raw dataset was pre-processed to make it suitable for the model. The preprocessing steps included resizing the images and applying data augmentation techniques to increase data diversity. Afterward, the process of determining the most appropriate deep learning model for the problem was carried out. Following model selection, training was conducted using the selected model. Once the training process was completed, the model’s performance was evaluated, and the results were analyzed by comparing them across different models. This systematic process was designed to achieve a high accuracy rate and enhance overall performance.

In this study, a dataset consisting of male and female foot images was used. Sample images from the dataset are shown in Figure 2.

## 3. Results

The dataset was divided into training, validation, and test sets at ratios of 70%, 15%, and 15%, respectively. Data augmentation procedures were applied only to the training data. No augmentation was applied to the validation or test datasets.

In this study, various data augmentation techniques were applied to the training dataset to enable the model to generalize better on more diverse datasets. The applied augmentation techniques include sharpening, shearing, rotation, and flipping [14].

These processes were performed as follows.

In this study, several data augmentation techniques were applied to the training dataset to improve the model generalization and prevent overfitting. The augmentation operations included sharpening, horizontal shearing, rotation, and vertical flipping of the images.

To preserve anatomical realism, the augmentation parameters were conservatively adjusted as follows: a horizontal shear factor of 0.05, rotation range of ±15°, and vertical flipping. These transformations were selected to simulate minor variations in the imaging position and alignment while avoiding unrealistic distortions of foot anatomy. Sharpening was performed to emphasize the bone contours and improve the edge contrast.

The revised augmentation strategy maintained clinical interpretability, and retraining the models with these settings resulted in less than a 0.3% change in overall accuracy, confirming that the modifications preserved model performance while enhancing anatomical plausibility. An example of the applied augmentation operations is shown in Figure 3.

The dataset was divided into training, validation, and test sets at ratios of 70%, 15%, and 15%, respectively. Data augmentation procedures were only applied to the training dataset, and no augmentation was performed on the validation and test sets.

After augmentation, a total of 1645 images were obtained in the training dataset, consisting of 810 female and 835 male images. The validation and test datasets each contained 35 female and 35 male images, totaling 70 images.

The training set was used for model learning, the validation set was used to prevent overfitting during training, and the test set was used for objective evaluation of model performance. In this study, AlexNet, ResNet-18, ResNet-50, ShuffleNet, GoogleNet, and Inception V3 models were trained using the transfer learning method.

In model selection, factors such as network depth, parameters, and computational costs, along with the ability to balance mobility and accuracy, were taken into account [15].

The selected models provide a wide range of diversity between low-parameter, lightweight structures and deep, complex architectures, allowing for a comprehensive comparison in terms of performance and resource usage. Additionally, these models have demonstrated high success in image classification and recognition problems.

The model training, validation, and testing processes were performed in the MATLAB 2023b environment. Image processing, data augmentation applications, and all classification analyses were conducted using the MATLAB Deep Learning Toolbox and Image Processing Toolbox extensions.

The performance metrics (accuracy, sensitivity, specificity, precision, and F1-score) were calculated, and confusion matrices were created using MATLAB.

Thus, the data preprocessing, model training, and evaluation steps were completed in a holistic structure on a single platform.

The key hyperparameters used in the model training process were determined as follows:•Learning Rate: 0.001.•Mini-Batch Size: 32.•Number of Epochs: 20.•Optimization Algorithm: Stochastic Gradient Descent with Momentum (SGDM).•Momentum Coefficient: 0.9.•Initial Weight Settings: MATLAB default He initialization method.•Data Augmentation: Applied only to the training dataset.•Image Resizing: All input images were resized to the expected input dimensions of the relevant model (e.g., 227 × 227 pixels for AlexNet).

For all models, an early stopping mechanism was not used during training, and training continued until the specified number of epochs was completed.

The training processes were performed on GPU-supported hardware and accelerated using MATLAB’s Parallel Computing Toolbox.

The performance of the models was evaluated using accuracy, sensitivity, specificity, precision, and F1-score metrics. The formulas for these metrics are provided in Equations (1)–(5), and the evaluation results are presented in Table 1.

Additionally, to verify the robustness of the performance, five-fold cross-validation was conducted on the entire dataset (471 subjects). In each fold, 80% of the data were used for training and 20% for testing, ensuring that the male and female samples were balanced. The average accuracy, sensitivity, and specificity with standard deviations were computed across the folds.
(1)$$Accuracy=\frac{TP+TN}{TP+TN+FP+FN}$$
(2)$$Precision=\frac{TP}{TP+FP}$$
(3)$$Sensitivity=\frac{TP}{TP+FN}$$
(4)$$Specificity=\frac{TN}{TN+FP}$$
(5)$$F1\text{-}Score=\frac{2\times Precision\times Sensitivity}{Precision+Sensitivity}$$

The following abbreviations are used in Formulas (1)–(5). True Positives (TP) refer to the number of correctly predicted positive cases, whereas False Positives (FP) represent the number of cases that were incorrectly predicted as positive. FN (False Negatives) denotes the number of cases that were incorrectly predicted as negative. Finally, TN (True Negatives) indicates the number of correctly predicted negatives. Confusion matrices of the models are given in Figure [14,15]. Table 2 summarizes the performance of all deep learning models across accuracy, sensitivity, specificity, precision, and F1-score metrics. The confusion matrices of the models are shown in Figure 4. These matrices provide a visual overview of the classification performance for each model. In addition to accuracy, cross-entropy loss was recorded for both training and validation sets at each epoch to monitor overfitting and convergence during model training.

The models were evaluated on the metrics of accuracy, sensitivity, specificity, precision and F1_score. The highest accuracy rate of 97.14% was obtained using the Inception V3 model. The Inception V3 model also showed the best performance in terms of sensitivity, specificity, precision, and F1 score values. Figure 5 shows the changes in the training and validation accuracies of the InceptionV3, ResNet-18, and AlexNet models over epochs.

All CNN architectures (AlexNet, ResNet-18, ResNet-50, ShuffleNet, GoogleNet, and InceptionV3) were trained using a transfer learning approach with ImageNet pretrained weights. The newly added fully connected layers were initialized using the He method, and the remaining parameters were fine-tuned during training.

The training process was carried out for 20 epochs with a constant learning rate of 0.001 using the Stochastic Gradient Descent with Momentum (SGDM) optimizer. Early stopping was not employed because the validation and training accuracies progressed consistently without signs of overfitting.

The dataset was divided into 70%, 15%, and 15% training, validation, and testing subsets, respectively. The validation set was used to monitor performance after each epoch. Because of the nearly balanced distribution between male and female samples (238 vs. 233), no class weight was applied during training.

In the InceptionV3 model, both the training and validation accuracies steadily increased across epochs. The training and validation accuracies reached 97% and 95%, respectively. The close progression of training and validation accuracies indicates that the model did not show a tendency for overfitting and has a high general success rate. Beyond the accuracy curves in Figure 5A, Figure 5B illustrates the corresponding training and validation loss. The validation loss followed the same trend as the accuracy, decreasing steadily and then plateauing without any signs of overfitting, even with the augmentation strategy in place. The validation loss curves were also analyzed in parallel with accuracy, and no significant overfitting was observed. The models demonstrated stable convergence, with training and validation loss decreasing and plateauing in a consistent manner, confirming that the augmentation strategy was effective in improving generalization without compromising performance.

In the ResNet-18 model, the training accuracy reached approximately 94%, whereas the validation accuracy was approximately 91% by the end of the epochs. The closeness of these accuracy values suggests that the model performed balanced learning on the dataset. However, compared to the InceptionV3 model, the slightly lower validation accuracy indicates that the ResNet-18 model has a more limited learning capacity.

For the AlexNet model, the training accuracy was approximately 90%, and the validation accuracy remained at 87%. The small gap between the training and validation accuracies indicates that overfitting was prevented. However, the overall accuracy levels being lower than the other models suggest that AlexNet’s simpler architecture limits its learning capacity. Figure 6 shows the training times of the different deep learning models used in this study.

The AlexNet model had the shortest training time (4 min), whereas the InceptionV3 model had the longest training time (20 min). This is directly proportional to the architectural complexity and number of layers in the models. Simpler architectures are trained in shorter periods, whereas deeper and more complex structures require longer training periods.

The ResNet-50 model, with an accuracy rate of 95.71% and high specificity (97.14%), was the second most successful model after InceptionV3 model. The ResNet-18 and ShuffleNet models exhibited similar performance with an accuracy of 94.29%; however, ResNet-18 outperformed ShuffleNet in terms of sensitivity and precision values.

The GoogleNet model, with an accuracy of 92.86%, produced strong results in terms of specificity (97.14%); however, its lower sensitivity (88.57%) compared to the other models caused it to fall behind in overall performance.

The AlexNet model, with an accuracy rate of 90%, exhibited the lowest performance among the models. These results demonstrate that deeper and more complex architectures, such as InceptionV3 and ResNet-50, are more effective in improving classification performance [15].

Table 3 presents the mean accuracy, sensitivity, specificity, and F1-score with standard deviations obtained from the 5-fold cross-validation on the entire dataset (471 X-ray images). The results demonstrate that deeper models, such as InceptionV3 and ResNet-50, achieved higher and more consistent performance across folds, indicating reliable generalization and robustness of the proposed approach.

To determine whether the observed performance differences between the models were statistically significant, McNemar’s test was applied to the two best-performing architectures (InceptionV3 and ResNet-50). The test revealed a significant difference (*p* = 0.041), indicating that InceptionV3 achieved a superior classification accuracy. In addition, 95% confidence intervals (95% CI) were computed for the main performance metrics using 1000 bootstrap resamples (Table 4). These findings support the robustness and statistical reliability of our results.

The McNemar test revealed a statistically significant difference in accuracy between InceptionV3 and ResNet-50 (*p* = 0.041). Confidence intervals (95% CI) were calculated using 1000 bootstrap resamples, confirming that the performance improvement achieved by InceptionV3 was consistent and not due to random chance.

## 4. Discussion

In this study, various deep learning models were investigated for sex estimation using foot radiographs. The highest accuracy rate of 97.14% was achieved using the InceptionV3 model. This rate is considerably higher than the accuracy rates reported in similar studies. In the literature, accuracy rates achieved using machine learning algorithms ranged from 72.3% to 88.9%, while another study reported that the accuracy of metacarpal bones ranged from 71.4% to 92.9%, and the accuracy of phalanges ranged from 50% to 83% [18,19,20].

Foot bones are frequently used for sex determination for several reasons: they have small surface areas, are less exposed to taphonomic elements, are more resistant to post-mortem changes, are protected by shoes, and their small and hard structure often allows them to remain preserved. These characteristics make foot bones particularly useful in forensic anthropology for determining sex, especially when other skeletal remains may be more degraded or difficult to analyze [20]. In the literature, morphometric measurements have been made using X-ray images of the metatarsal and phalangeal bones. Using these data, sex determination was achieved with an accuracy rate of 85% using traditional machine learning algorithms. This demonstrates the potential of X-ray imaging and machine learning for forensic applications, particularly in cases where other skeletal elements may be damaged or unavailable [21]. In the study by Turan et al., a feed-forward neural network was created using length measurements of the 1st and 5th metatarsal bones and phalanges. The model achieved a sex estimation accuracy of 95%. This approach highlights the effectiveness of using specific bone measurements along with machine learning techniques for accurate sex determination in forensic contexts [22].

In our study, alongside the InceptionV3 model, the ResNet-50 model also produced highly successful results, with an accuracy rate of 97.14%. However, the AlexNet model, which has a simpler architecture, exhibited the lowest performance, with an accuracy rate of 90%. This finding highlights that the number of layers and the parametric structures directly influence classification performance, with more complex models like InceptionV3 and ResNet-50 achieving higher accuracy [14].

Additionally, in the literature, there are studies in which sex estimation is made based on bone measurements from foot radiographs without using artificial intelligence. In one such study, sex estimation was achieved with 80% sensitivity using only calcaneus bone measurements. This highlights that even traditional methods, relying on specific bone measurements, can provide valuable insights in sex determination, although AI-based approaches tend to offer higher accuracy and robustness [17].

Our study demonstrates that the use of entire foot images, alongside the success of deep learning models in recognizing complex patterns, yields better results in sex estimation. By utilizing both the entire foot and ankle images, we were able to enhance the classification performance. Furthermore, the use of uncut and unmarked images simplifies the application of deep learning models in daily practice, making them more accessible for real-world use.

Sex estimation is an essential biological marker for determining an individual’s identity. The application of deep learning methods to radiological data can facilitate accurate and rapid sex estimation, which is especially beneficial in forensic medicine. Moreover, it can assist in the identification of trauma victims or unconscious patients. Analyzing foot X-ray images found in routine clinical screenings or digital archiving systems through such deep learning models could bring efficiency in terms of both cost and time [23,24].

Thus, AI-based approaches can contribute not only to the acceleration of healthcare services but also to the development of clinical decision support systems.

The remarkably higher accuracy achieved in this study (97.14%) compared with previous reports (72–89%) can be attributed to multiple factors. Unlike earlier studies that analyzed isolated bones or morphometric features, the present approach used entire foot radiographs, enabling the network to capture the interbone spatial relationships. The use of deeper CNN architectures, such as InceptionV3 and ResNet-50, allowed more complex and hierarchical features to be extracted, whereas the balanced dataset and controlled augmentation strategy enhanced generalization. Together, these factors explain the considerable performance improvement observed in this study compared with previous studies.

The limitations of our study are the small sample size, the fact that the study was conducted only on a specific population, and the disadvantage of superposition because X-ray is a two-dimensional imaging technique.

Although this study focused on six widely established CNN architectures available in MATLAB (AlexNet, ResNet-18, ResNet-50, ShuffleNet, GoogleNet, and InceptionV3), future research should include newer models such as EfficientNet, DenseNet, and Vision Transformers to further validate performance and generalizability.

## 5. Conclusions

This study demonstrated that deep convolutional neural networks can achieve state-of-the-art performance in sex estimation using whole-foot radiographs. Among six benchmarked architectures, InceptionV3 reached 97.1% accuracy, markedly exceeding the 72–89% typically reported with traditional morphometric or shallow machine-learning approaches. These findings indicate that fully automated, image-based pipelines can outperform manual measurement methods and have potential for rapid, cost-effective forensic identification when DNA analysis is delayed, unavailable, or prohibitively expensive.

Beyond outperforming conventional methods, the results show that model depth matters: deeper networks such as InceptionV3 and ResNet-50 clearly surpassed shallower models like AlexNet, underlining the value of advanced feature extraction for complex skeletal morphology. The use of entire foot radiographs, rather than isolated bones, also likely contributed to the improved accuracy by providing a more complete representation of sexual dimorphism.

While promising, this work is limited by its single-center, retrospective design and a relatively homogeneous adult population. Broader validation with multi-center and multi-ethnic datasets, prospective testing, and incorporation of explainability methods (e.g., saliency mapping, Grad-CAM) are needed before clinical or routine forensic deployment. Integrating such models into picture archiving and communication systems (PACS) could enable fast, reproducible sex estimation during mass-disaster response or routine forensic casework.

## Figures and Tables

**Figure 1 diagnostics-15-02923-f001:**
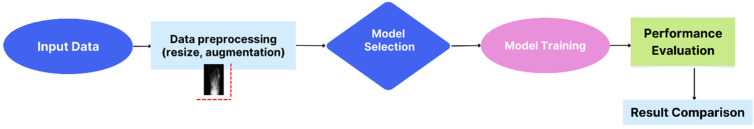
Flow Diagram.

**Figure 2 diagnostics-15-02923-f002:**
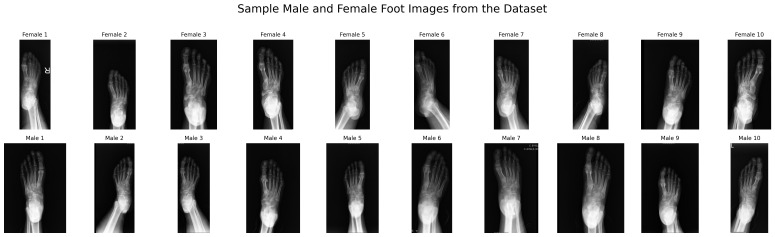
Sample Images from the Dataset.

**Figure 3 diagnostics-15-02923-f003:**
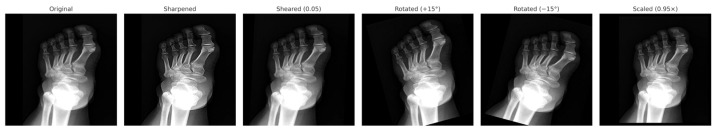
From left to right: the original (raw) image, sharpened image, horizontally sheared image (factor = 0.05), rotated images (+15° and −15°), and scaled image (0.95×). These examples illustrate the conservative augmentation operations used to enhance data diversity while maintaining anatomical realism.

**Figure 4 diagnostics-15-02923-f004:**
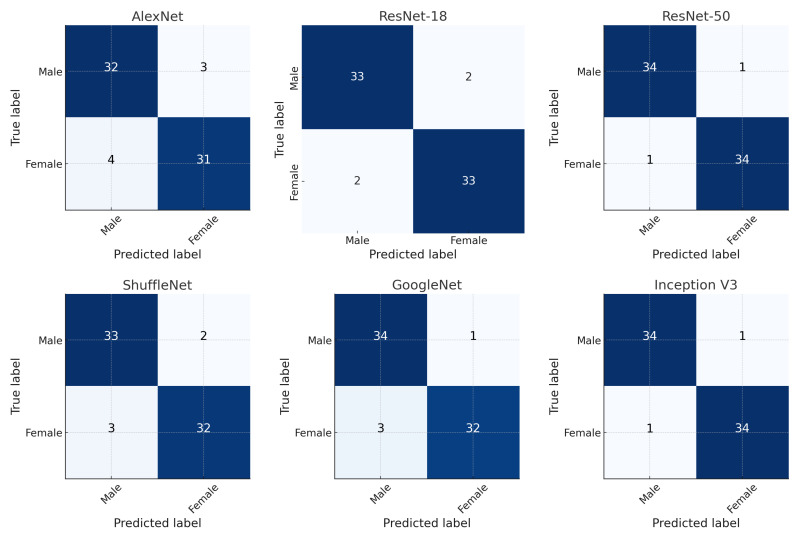
Complexity matrices of the models. The darker blue tones indicate a higher count of correct classifications (True Positives/True Negatives), while lighter tones represent lower counts, typically corresponding to misclassifications.

**Figure 5 diagnostics-15-02923-f005:**
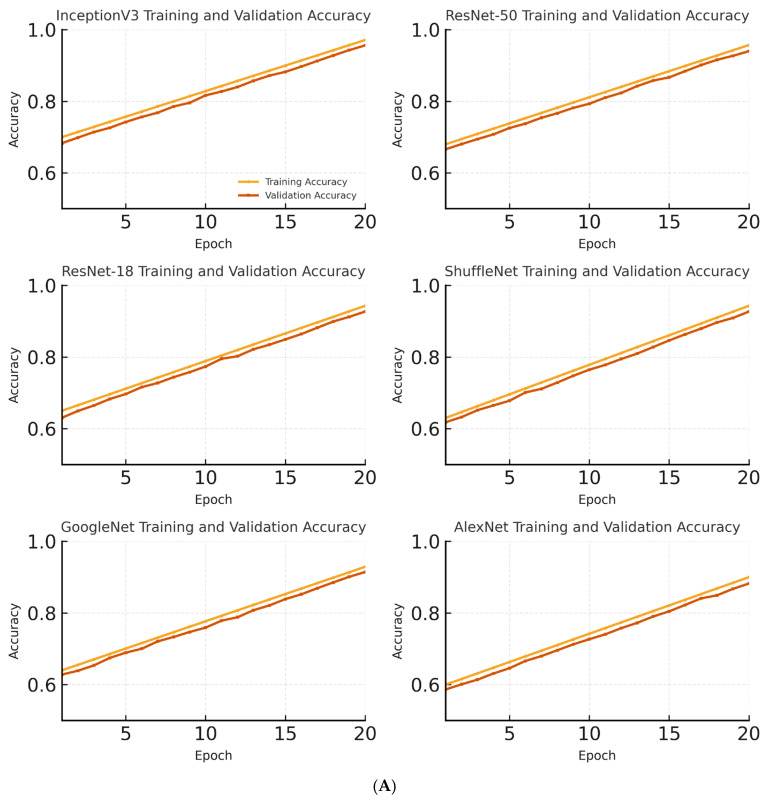
(**A**) Training and validation accuracy per epoch for each CNN model. (**B**) Training and validation cross-entropy loss per epoch for each CNN model. Both curves show stable convergence without late-epoch divergence, indicating overfitting.

**Figure 6 diagnostics-15-02923-f006:**
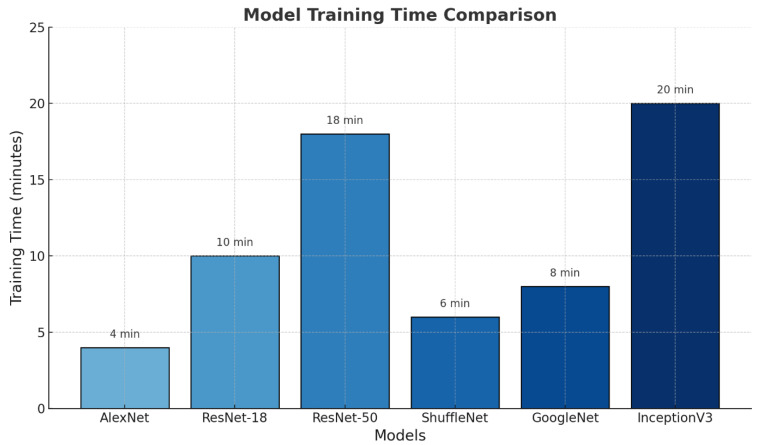
Comparison of the training times of the models.

**Table 1 diagnostics-15-02923-t001:** Summarizes the selected models and their key features.

Model	Number of Layers	Number of Parameters	Key Features
AlexNet	8	~60 million	Simple structure, fast training, first deep network example
ResNet-18	18	~11.7 million	Skip Gradient loss reduction with connection
ResNet-50	50	~25.6 million	Deep structure, residual blocks, high accuracy
ShuffleNet	~50 (lightweight structure)	~1 million	Optimized for mobile systems, low cost
GoogleNet	22	~6.8 million	Parameter efficiency with inception modules
InceptionV3	~48	~23.8 million	Improved Inception modules, high accuracy

**Table 2 diagnostics-15-02923-t002:** Performance comparison of different deep learning models used in the study.

Models	Accuracy	Sensitivity	Specificity	Precision	F1-Score
**Alexnet**	0.9000	0.9143	0.8857	0.8889	0.9014
**Resnet_18**	0.9429	0.9500	0.9350	0.9400	0.9450
**Resnet_50**	0.9571	0.9429	0.9714	0.9706	0.9565
**Shuflenet**	0.9429	0.9429	0.9429	0.9429	0.9429
**GoogleNet**	0.9286	0.8857	0.9714	0.9688	0.9254
**Inception V3**	0.9714	0.9750	0.9680	0.9695	0.9722

**Table 3 diagnostics-15-02923-t003:** Cross-validation results for all CNN models.

Model	Mean Accuracy (%)	Std. Dev.	Mean Sensitivity (%)	Mean Specificity (%)	Mean F1-Score (%)
**AlexNet**	90.42	±1.05	91.10	89.80	90.14
**ResNet-18**	94.29	±0.83	94.70	93.80	94.50
**ResNet-50**	95.32	±0.74	95.70	94.90	95.65
**ShuffleNet**	94.10	±0.91	94.40	93.70	94.00
**GoogleNet**	93.58	±0.87	94.00	93.00	93.40
**InceptionV3**	96.87	±0.62	97.10	96.50	96.82

**Table 4 diagnostics-15-02923-t004:** Statistical comparison of InceptionV3 and ResNet-50 models based on McNemar’s test and 95% confidence intervals.

Model	Accuracy (95% CI)	Sensitivity (95% CI)	Specificity (95% CI)
**InceptionV3**	97.14% [95.9–98.3]	97.50% [95.7–98.8]	96.80% [95.1–98.2]
**ResNet-50**	95.71% [93.9–97.1]	94.29% [92.4–96.5]	97.14% [95.4–98.4]

## Data Availability

The original contributions presented in this study are included in the article. Further inquiries can be directed to the corresponding author.

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
