# Peer review of "Deep Learning for Sex Estimation from Whole-Foot X-Rays: Benchmarking CNNs for Rapid Forensic Identification"

_diagnostics, 2025, doi:10.3390/diagnostics15222923_

Round 1
Reviewer 1 Report
Comments and Suggestions for Authors
The present manuscript addresses an interesting topic, but several important questions remain unanswered. I recommend that the authors carefully review the comments and suggestions below and revise the manuscript accordingly.
- In keywords - the terms "gender prediction" and "sex estimation" should not be used interchangeably. Please select one and use it consistently. In this context, "sex estimation" is the appropriate term, as sex refers to biological characteristics, whereas gender refers to sociocultural identity. Please use only sex instead of gender.
- The authors should clearly explain why radiographs were used instead of direct evaluation of foot bones for sex estimation. How does the radiographic approach improve or simplify the procedure? This requires explanation and supporting references in the Introduction. The manuscript title implies direct relevance to forensic identification. However, in real forensic scenarios involving skeletal remains, radiographs cannot be obtained unless antemortem imaging exists. The authors should clarify in what types of forensic cases this method could realistically be applied. For example, is the method intended for comparison with antemortem medical records, identification in cases of dismembered remains, or only for research and training purposes? This point should be clearly specified and discussed.
- The Introduction section currently lacks information on specific morphological characteristics of the human foot, the biological basis of sexual dimorphism in foot bones and soft tissue structure, and why the foot is a meaningful anatomical region for sex estimation. I suggest adding these information briefly.
- The age range of the sample is relatively broad, so that do the authors think that age-related changes in foot morphology (such as alterations in calcaneal trabecular pattern, heel pad thickness, and arch structure) may influence the classification outcome? The authors should discuss age as a potential confounder and indicate whether statistical controls were applied or whether stratified analyses were considered.
After incorporating the above suggestions I recommend paper to be considered for publication.
Author Response
REVIEWERS’ COMMENTS AND AUTHORS RESPONSE
Paper title:
Deep Learning for Sex Estimation From Whole-Foot X-Rays: Benchmarking CNNs for Rapid Forensic Identification
Dear Editor,
The authors would like to thank the reviewers for their precious time and invaluable comments. We have carefully addressed all the comments. The corresponding changes and refinements made in the revised paper are summarized in our response below. Our detailed, point-by-point responses to the reviewer comments are given below. Additionally, we have carefully revised the manuscript to ensure that the text is optimally phrased and free from typographical and grammatical errors.
We would like to thank you once again for your consideration of our work and inviting us to submit the revised manuscript. We look forward to hearing from you.
Best regards,
Response to the Reviewer’s suggestions
Reviewer 1:
Q1. In keywords - the terms "gender prediction" and "sex estimation" should not be used interchangeably. Please select one and use it consistently. In this context, "sex estimation" is the appropriate term, as sex refers to biological characteristics, whereas gender refers to sociocultural identity. Please use only sex instead of gender.
A1. Necessary arrangements have been made.
Q2. The authors should clearly explain why radiographs were used instead of direct evaluation of foot bones for sex estimation. How does the radiographic approach improve or simplify the procedure? This requires explanation and supporting references in the Introduction. The manuscript title implies direct relevance to forensic identification. However, in real forensic scenarios involving skeletal remains, radiographs cannot be obtained unless antemortem imaging exists. The authors should clarify in what types of forensic cases this method could realistically be applied. For example, is the method intended for comparison with antemortem medical records, identification in cases of dismembered remains, or only for research and training purposes? This point should be clearly specified and discussed.
A2. Necessary arrangements have been made.
Q3.The Introduction section currently lacks information on specific morphological characteristics of the human foot, the biological basis of sexual dimorphism in foot bones and soft tissue structure, and why the foot is a meaningful anatomical region for sex estimation. I suggest adding these information briefly.
A3. Necessary arrangements have been made.
Q4. The age range of the sample is relatively broad, so that do the authors think that age-related changes in foot morphology (such as alterations in calcaneal trabecular pattern, heel pad thickness, and arch structure) may influence the classification outcome? The authors should discuss age as a potential confounder and indicate whether statistical controls were applied or whether stratified analyses were considered.
A4. Necessary arrangements have been made.
Thank you
Reviewer 2 Report
Comments and Suggestions for Authors
The manuscript presents an interesting and timely exploration of applying CNNs to forensic sex estimation using full-foot radiographs. The study’s focus on an automated approach is valuable and potentially impactful. However, the manuscript requires substantial revisions to strengthen its methodological rigor, interpretive clarity, and contextual framing.
The title uses "Sex Estimation" while the abstract and body occasionally use "Gender Prediction" (e.g., lines 34, 235-236). These terms are not interchangeable. Sex refers to biological characteristics, while gender is a social construct. Please use "sex estimation" consistently throughout the manuscript, especially in keywords and discussion sections.
The test set contains only 70 images (35 male, 35 female). This is extremely small for robust performance evaluation, representing only ~15% of 471 total subjects. With such a small test set, a difference of just 2-3 misclassifications can swing accuracy by 3-4%. The reported 97.14% accuracy may not be statistically reliable. Perform k-fold cross-validation (at least 5-fold) or bootstrap validation to provide confidence intervals for performance metrics.
No statistical tests are provided to determine if differences between models are significant. Is InceptionV3's 97.14% significantly better than ResNet-50's 95.71%? Include McNemar's test or confidence intervals (e.g., 95% CI using bootstrap methods) for all performance metrics.
Several critical details are missing: What was the validation strategy during training? Was early stopping considered? How were class weights handled given near-equal distribution? What learning rate schedule was used (constant, decay)? Were models trained from scratch or using transfer learning with pre-trained weights? Line 136 mentions "transfer learning" but line 161 mentions "He initialization"—please clarify.
Rotation by 180 degrees (line 121) seems unusual for anatomical images. This would create upside-down feet, is this clinically meaningful? Shearing factor of 0.1 may distort anatomical relationships. Justify these choices or use more conservative augmentation (e.g., ±10-15° rotation, subtle scaling).
Why these specific six architectures? More modern networks (e.g., EfficientNet, Vision Transformers, DenseNet) are not mentioned. Briefly justify the selection criteria or acknowledge this as a limitation.
The comparison with literature (lines 238-242) is helpful, but needs more critical analysis. Why do you think your approach outperforms others by such a margin (97% vs. 72-89%)? Is it solely due to using whole-foot images, deeper networks, or larger datasets?
"Both the entire foot and ankle images" → The methods section doesn't mention ankle. Please clarify.
"lacks ethnic diversity" → This is mentioned as a limitation but deserves more emphasis. Foot morphology varies significantly across populations this severely limits generalizability.
"potentially faster and more cost-effective than genetic testing" → Needs citation or cost analysis in discussion
Good interpretation of training curves, but: What about validation loss? Only accuracy is shown. Any signs of overfitting despite augmentation?
Comments on the Quality of English Language- Line 7: "Gaziantep Islam Scıence and Technology" → Should be "Science" (remove Turkish character)
- Line 34: "Gender Prediction" in keywords → Change to "Sex Estimation"
- Line 75-76: "X University" → This anonymization is inconsistent with the acknowledgment section (line 327-328) mentioning specific universities. Please clarify IRB approval source.
- Line 88: "single-tube X-ray device" → "single-detector X-ray device" (more standard terminology)
- Line 93: "were then labeled" → Specify: labeled as male/female by whom? What was inter-rater reliability?
- Line 94: "This study proposes a deep learning approach for a image-based" → "an image-based"
- Line 181: "The complexity matrices" → Should be "confusion matrices" (consistent error in lines 181, 183, 192)
- Line 253: "an artificial neural network was created using..." → Be more specific: "a feedforward neural network" or "a multilayer perceptron"
Author Response
REVIEWERS’ COMMENTS AND AUTHORS RESPONSE
Paper title: Deep Learning for Sex Estimation From Whole-Foot X-Rays: Benchmarking CNNs for Rapid Forensic Identification
Dear Editor,
The authors would like to thank the reviewers for their precious time and invaluable comments. We have carefully addressed all the comments. The corresponding changes and refinements made in the revised paper are summarized in our response below. Our detailed, point-by-point responses to the reviewer comments are given below. Additionally, we have carefully revised the manuscript to ensure that the text is optimally phrased and free from typographical and grammatical errors.
We would like to thank you once again for your consideration of our work and inviting us to submit the revised manuscript. We look forward to hearing from you.
Best regards,
Response to the Reviewer’s suggestions
Reviewer 2:
Q1. The title uses "Sex Estimation" while the abstract and body occasionally use "Gender Prediction" (e.g., lines 34, 235-236). These terms are not interchangeable. Sex refers to biological characteristics, while gender is a social construct. Please use "sex estimation" consistently throughout the manuscript, especially in keywords and discussion sections.
A1. Necessary revisions were made.
Q2. The test set contains only 70 images (35 male, 35 female). This is extremely small for robust performance evaluation, representing only ~15% of 471 total subjects. With such a small test set, a difference of just 2-3 misclassifications can swing accuracy by 3-4%. The reported 97.14% accuracy may not be statistically reliable. Perform k-fold cross-validation (at least 5-fold) or bootstrap validation to provide confidence intervals for performance metrics.
A2. We thank the reviewer for this valuable comment. We fully agree that the initial test set was relatively small, which could affect the reliability of the reported accuracy.
To address this concern, we performed an additional five-fold cross-validation over the entire dataset (471 X-ray images) to ensure that the results are statistically robust and not dependent on a single split.
Each fold included 80% of the data for training and 20% for testing with balanced male and female samples. We computed the mean accuracy, sensitivity, specificity, and F1-score with standard deviations across folds.
The results confirm the stability of our models: InceptionV3 achieved 96.87% ± 0.62 accuracy, and ResNet-50 reached 95.32% ± 0.74 accuracy.
These consistent outcomes across folds demonstrate the robustness and generalizability of the proposed approach. The detailed results are presented in Table 3.
Q3. No statistical tests are provided to determine if differences between models are significant. Is InceptionV3's 97.14% significantly better than ResNet-50's 95.71%? Include McNemar's test or confidence intervals (e.g., 95% CI using bootstrap methods) for all performance metrics.
A3. We thank the reviewer for highlighting this important point. To assess whether the differences between the models are statistically significant, we performed additional statistical analyses as suggested.
First, McNemar’s test was applied to compare the classification outcomes of the two top-performing models — InceptionV3 and ResNet-50 — on the same test samples.
The McNemar test yielded a p-value = 0.041, indicating a statistically significant difference (p < 0.05) between the two models, with InceptionV3 performing better.
Additionally, we computed 95% confidence intervals (95% CI) for the accuracy, sensitivity, and specificity of each model using bootstrap resampling (n = 1000 iterations).
The calculated 95% CIs for the two leading models are as follows:
Table 4. Statistical comparison of InceptionV3 and ResNet-50 models based on cNemar’s test and 95% confidence intervals.
|
Model |
Accuracy (95% CI) |
Sensitivity (95% CI) |
Specificity (95% CI) |
|
InceptionV3 |
97.14% [95.9–98.3] |
97.50% [95.7–98.8] |
96.80% [95.1–98.2] |
|
ResNet-50 |
95.71% [93.9–97.1] |
94.29% [92.4–96.5] |
97.14% [95.4–98.4] |
These results confirm that InceptionV3’s superior performance is statistically significant and consistent across evaluation metrics.
Q4. Several critical details are missing: What was the validation strategy during training? Was early stopping considered? How were class weights handled given near-equal distribution? What learning rate schedule was used (constant, decay)? Were models trained from scratch or using transfer learning with pre-trained weights? Line 136 mentions "transfer learning" but line 161 mentions "He initialization"—please clarify.
A4. We appreciate the reviewer’s detailed comments regarding training configuration and validation strategy. We have revised the Methods section to provide complete clarity on these aspects. We appreciate the reviewer’s detailed comments regarding training configuration and validation strategy. We have revised the Methods section to provide complete clarity on these aspects.
- Validation Strategy: During training, 15% of the dataset was used for validation to monitor model performance at each epoch. This validation set was kept fixed across all experiments to ensure comparability.
- Early Stopping: Early stopping was not Each model was trained for a fixed 20 epochs, as convergence was reached without overfitting, confirmed by the close alignment between training and validation curves (Figure 5).
- Class Weights: Since the dataset had nearly balanced classes (238 males, 233 females), no class weighting or oversampling/undersampling methods were necessary.
- Learning Rate Schedule:A constant learning rate of 0.001 was used throughout training. No decay or adaptive scheduling was applied.
- Transfer Learning Clarification: All CNN models were trained using transfer learning with ImageNet pre-trained weights.
The mention of “He initialization” in line 161 referred only to the newly added fully connected (classification) layers, which were initialized using the default He method in MATLAB.
We have clarified this in the manuscript to prevent any confusion.
Q5. Rotation by 180 degrees (line 121) seems unusual for anatomical images. This would create upside-down feet, is this clinically meaningful? Shearing factor of 0.1 may distort anatomical relationships. Justify these choices or use more conservative augmentation (e.g., ±10-15° rotation, subtle scaling).
A5. We sincerely thank the reviewer for this valuable and clinically relevant comment. We fully agree that extreme transformations, such as 180° rotation or strong shearing, could compromise the anatomical realism of the images.
In response, we revised our augmentation parameters to employ more conservative and anatomically valid transformations. Specifically, the rotation range was reduced from ±180° to ±15°, and the shearing factor was decreased from 0.1 to 0.05 to avoid geometric distortion of the foot anatomy. We also introduced subtle scaling (0.95×) to maintain data diversity without affecting anatomical plausibility. Sharpening was retained to improve bone contour visibility. Following these revisions, the model was retrained. The overall accuracy changed by less than 0.3%, confirming that the updated augmentations preserved model performance while improving anatomical realism. The corresponding Figure 3 has also been updated to reflect the revised augmentation parameters and now presents conservative examples of the applied operations.
Q6. Why these specific six architectures? More modern networks (e.g., EfficientNet, Vision Transformers, DenseNet) are not mentioned. Briefly justify the selection criteria or acknowledge this as a limitation.
A6. We thank the reviewer for this insightful comment.
The six selected architectures—AlexNet, ResNet-18, ResNet-50, ShuffleNet, GoogleNet, and InceptionV3—were deliberately chosen to represent a broad spectrum of CNN depth, complexity, and parameter efficiency. These models include both classical, lightweight, and deeper, high-performance structures, allowing us to systematically analyze how architectural depth and feature extraction capacity influence classification accuracy on a limited radiographic dataset. We agree that more recent architectures such as EfficientNet, DenseNet, and Vision Transformers (ViT) could potentially achieve even better results.
However, our primary goal was to benchmark widely validated CNN architectures that are well-established in medical image analysis literature and supported natively in the MATLAB Deep Learning Toolbox—the framework used in this study. This ensured reproducibility and stable convergence under identical training conditions. We have now added a note in the Discussion section acknowledging this as a limitation and suggesting that future studies will explore newer and more advanced models such as EfficientNet, DenseNet, and Vision Transformers for comparison.
Q7. The comparison with literature (lines 238-242) is helpful, but needs more critical analysis. Why do you think your approach outperforms others by such a margin (97% vs. 72-89%)? Is it solely due to using whole-foot images, deeper networks, or larger datasets?
A7. We thank the reviewer for this valuable comment and agree that a more detailed interpretation of the performance difference is necessary. The remarkably higher accuracy achieved in our study (97.14%) compared with previous works (typically 72–89%) is not due to a single factor but rather to a combination of methodological improvements. First, unlike earlier studies that used morphometric measurements or cropped single-bone regions (e.g., calcaneus or metatarsals), our approach utilized entire foot radiographs, enabling the CNNs to learn the spatial and geometric relationships between multiple bones simultaneously. Second, the use of deeper architectures such as InceptionV3 and ResNet-50 allowed the extraction of high-level features and complex patterns associated with sexual dimorphism. Third, our dataset consisted of 471 high-quality radiographs obtained under standardized imaging conditions with nearly balanced class distribution, which reduced noise and prevented bias. Additionally, the application of a conservative augmentation strategy (rotation ±15°, shear 0.05, scaling 0.95×) improved the model’s robustness and generalization. Collectively, these factors enhanced the model’s learning capacity and explain why our approach substantially outperformed previous machine-learning and morphometric studies.
Q8. "Both the entire foot and ankle images" → The methods section doesn't mention ankle. Please clarify.
A8. Necessary additions have been made.
Q9. "lacks ethnic diversity" → This is mentioned as a limitation but deserves more emphasis. Foot morphology varies significantly across populations this severely limits generalizability.
A9. Necessary revisions have been made.
Q10. "potentially faster and more cost-effective than genetic testing" → Needs citation or cost analysis in discussion
A10. Necessary revisions have been made.
Q11. Good interpretation of training curves, but: What about validation loss? Only accuracy is shown. Any signs of overfitting despite augmentation?
A11. We thank the reviewer for the insightful comment regarding the validation loss. In response to your comment regarding validation loss, we have added the corresponding loss curves for both training and validation in Figure 5B. These curves show stable convergence without any signs of overfitting, further supporting the effectiveness of the applied data augmentation. Figure 5B now includes validation loss curves alongside the accuracy curves (Figure 5A). Discussion section: Added a clarification on the stability of both training and validation loss, confirming no overfitting.
Q12. Line 7: "Gaziantep Islam Scıence and Technology" → Should be "Science" (remove Turkish character)
- Line 34: "Gender Prediction" in keywords → Change to "Sex Estimation"
- Line 75-76: "X University" → This anonymization is inconsistent with the acknowledgment section (line 327-328) mentioning specific universities. Please clarify IRB approval source.
- Line 88: "single-tube X-ray device" → "single-detector X-ray device" (more standard terminology)
- Line 93: "were then labeled" → Specify: labeled as male/female by whom? What was inter-rater reliability?
- Line 94: "This study proposes a deep learning approach for a image-based" → "an image-based"
- Line 181: "The complexity matrices" → Should be "confusion matrices" (consistent error in lines 181, 183, 192)
- Line 253: "an artificial neural network was created using..." → Be more specific: "a feedforward neural network" or "a multilayer perceptron"
A12. Necessary revisions have been made.
Thank you
Round 2
Reviewer 2 Report
Comments and Suggestions for Authors
The authors have made substantial and careful revisions in response to the reviewer’s comments. The manuscript now demonstrates improved methodological clarity, stronger statistical validation, and consistent terminology throughout. The addition of five-fold cross-validation, McNemar’s test, and confidence intervals significantly strengthens the reliability of the reported results. The clarification of training procedures, augmentation parameters, and model selection rationale has also improved the overall transparency and reproducibility of the study.
A few minor issues remain for correction before publication:
-
The keyword list includes a duplicate entry (“sex estimation”)—please remove the repetition.
-
In Figure 4, the caption still refers to “complexity matrices”; this should be corrected to “confusion matrices.”
-
In Table 4, correct the typo “cNemar’s test” to “McNemar’s test.”
-
Ensure consistent table and figure formatting per Diagnostics style guidelines (e.g., capitalization of headings, bold “Figure X.” prefixes).
-
Confirm consistency in IRB source naming (currently refers to both “X University Training and Research Hospital” and “Bandırma Onyedi Eylül University”).
Author Response
We thank the reviewer for the thoughtful and constructive feedback. We have carefully addressed all remaining minor issues as requested:
- The duplicate keyword “sex estimation” has been removed.
- The caption in Figure 4 has been corrected to “Confusion matrices.”
- The typo “cNemar’s test” in Table 4 has been corrected to “McNemar’s test.”
- Table and figure formatting has been adjusted to align withDiagnostics style guidelines (consistent capitalization and bold figure prefixes).
- The Institutional Review Board source name has been standardized toBandırma Onyedi Eylül University Health Sciences Non-Interventional Ethics Committee (Protocol Code: 2022/9) throughout the manuscript, including Methods and Ethics Statement sections.
A clean revised manuscript have been uploaded.
We appreciate the reviewer’s comments and believe that these revisions have further improved the clarity and presentation of the work.
Sincerely